# BioCRNpyler: Compiling chemical reaction networks from biomolecular parts in diverse contexts

**William Poole**[1]*, **Ayush Pandey**[2], **Andrey Shur**[3], **Zoltan A. Tuza**[4], **Richard M. Murray**[2]

**1** Computation and Neural Systems, California Institute of Technology, Pasadena, California, United States of America, **2** Control and Dynamical Systems, California Institute of Technology, Pasadena, California, United States of America, **3** Bioengineering, California Institute of Technology, Pasadena, California, United States of America, **4** Bioengineering, Imperial College London, London, England

* wpoole@caltech.edu

**Data Availability Statement:** BioCRNpyler source code and an extensive set of example notebooks, documentation, and tutorials are available in our GitHub repository: https://github.com/BuildACell/

## Abstract

Biochemical interactions in systems and synthetic biology are often modeled with chemical reaction networks (CRNs). CRNs provide a principled modeling environment capable of expressing a huge range of biochemical processes. In this paper, we present a software toolbox, written in Python, that compiles high-level design specifications represented using a modular library of biochemical parts, mechanisms, and contexts to CRN implementations. This compilation process offers four advantages. First, the building of the actual CRN representation is automatic and outputs Systems Biology Markup Language (SBML) models compatible with numerous simulators. Second, a library of modular biochemical components allows for different architectures and implementations of biochemical circuits to be represented succinctly with design choices propagated throughout the underlying CRN automatically. This prevents the often occurring mismatch between high-level designs and model dynamics. Third, high-level design specification can be embedded into diverse biomolecular environments, such as cell-free extracts and *in vivo* milieus. Finally, our software toolbox has a parameter database, which allows users to rapidly prototype large models using very few parameters which can be customized later. By using BioCRNpyler, users ranging from expert modelers to novice script-writers can easily build, manage, and explore sophisticated biochemical models using diverse biochemical implementations, environments, and modeling assumptions.

## Author summary

This paper describes a new software package BioCRNpyler (pronounced "Biocompiler") designed to support rapid development and exploration of mathematical models of biochemical networks and circuits by computational biologists, systems biologists, and synthetic biologists. BioCRNpyler allows its users to generate large complex models using very few lines of code in a way that is modular. To do this, BioCRNpyler uses a powerful

BioCRNpyler. All other data are available within the manuscript and its Supporting information files.

**Funding:** The authors WP and AP are partially supported by US National Science Foundation (CBET-1903477). AP was also supported by the Defense Advanced Research Projects Agency (Agreement HR0011-17-2-0008). AS was supported by the Institute for Collaborative Biotechnologies through cooperative agreement W911NF-19-2-0026 from the U.S. Army Research Office. The content of the information does not necessarily reflect the position or the policy of the Government, and no official endorsement should be inferred. The funders had no role in study design, data collection and analysis, decision to publish, or preparation of the manuscript.

**Competing interests:** The authors have declared that no competing interests exist.

new representation of biochemical circuits which defines their parts, underlying biochemical mechanisms, and chemical context independently. BioCRNpyler was developed as a Python scripting language designed to be accessible to beginning users as well as easily extendable and customizable for advanced users. Ultimately, we see Biocrnpyler being used to accelerate computer automated design of biochemical circuits and model driven hypothesis generation in biology.

This is a *PLOS Computational Biology* Software paper.

## 1 Introduction

Chemical reaction networks (CRNs) are the workhorse for modeling in systems and synthetic biology [1]. The power of CRNs lies in their expressivity; CRN models can range from physically realistic descriptions of individual molecules to coarse-grained idealizations of complex multi-step processes [2]. However, this expressivity comes at a cost. Choosing the right level of detail in a model is more an art than a science. The modeling process requires careful consideration of the desired use of the model, the available data to parameterize the model, and prioritization of certain aspects of modeling or analysis over others. Additionally, biological CRN models can be incredibly complex including dozens or even hundreds or thousands of species, reactions, and parameters [3]. Maintaining complex hand-built models is challenging and errors can quickly grow out of control for large models. Software tools can answer many of these challenges by automating and streamlining the model construction process.

Formally, a CRN is a set of species $S = \{S_i\}$ and reactions $R : \{I \xrightarrow{\rho(s;\theta)} O\}$ where $I$ and $O$ are multisets of species, $\rho$ is the rate function or propensity, $s$ is a vector of species' concentrations (or counts), and $\theta$ are rate parameters. Typically, CRNs are simulated using as ordinary differential equations (ODEs) and numerically integrated [2]. A stochastic semantics also allows CRNs to be simulated as continuous-time Markov chains [4]. Besides their prevalence in biological modeling, there is rich theoretical body of work related to CRNs from the mathematical [5], computer science [6], and physics communities [7]. Despite these theoretical foundations, many models are phenomenological in nature and lack mechanistic details of various biological processes. The challenge of constructing correct models is compounded by the difficulty in differentiating between correct and incorrect models based upon experimental data [8–10].

Due to CRNs' rich history and diverse applications, the available tools for a CRN modeler are vast and include: extensive software to generate, simulate, and analyze CRNs [11–14] as well as databases of models [15, 16], and many more. However, even though synthetic biologists have adopted a module and part-driven approach to their laboratory work [17], models are still typically built by hand on a case-by-case basis. Recognizing the fragile non-modular nature of hand built models, several synthetic biology design automation tools have been developed for specific purposes such as implementing transcription factor or integrase-based logic [18, 19]. These tools indicate a growing need for design and simulation automation in synthetic biology, as part and design libraries are expanded.

As the name would suggest, BioCRNpyler (pronounced bio-compiler) is a Python package that compiles CRNs from simple specifications of biological motifs and contexts. This package

is inspired by the molecular compilers developed by the DNA-strand displacement community and molecular programming communities which, broadly speaking, aim to compile models of DNA circuit implementations from simpler CRN specifications [20–22], rudimentary programming languages [23, 24], and abstract sequence specifications [25]. This body of work has demonstrated the utility of molecular circuit compilers and highlights that a single specification can be compiled into multiple molecular implementations which in turn can correspond to multiple CRN models at various levels of detail. For example, there are multiple DNA-strand implementations of catalysis [21, 22, 26, 27] and the interactions of the DNA strands involved in each of these implementations can be enumerated to generate different CRN models based upon the assumptions underlying enumeration algorithm [28]. Drawing from these inspirations, BioCRNpyler is a general-purpose CRN compiler capable of converting abstract specifications of biomolecular components into CRN models with full programmatic control over the compilation process. Importantly, BioCRNpyler is not a CRN simulator —models are saved in the Systems Biology Markup Language (SBML) [29] to be compatible with the user's simulator of choice.

There are many existing tools that provide some of the features present in BioCRNpyler. Antimony (part of the Tellerium software suite) provides an elegant high level language that is converted into SBML models [12, 30]. Systems Biology Open Language (SBOL) [31] is a format for sharing DNA-sequences with assigned functions and does not compile a CRN. Hierarchical SBML and supporting software [32] provide a file format which encapsulates CRNs as modular functions. The software package iBioSim [33, 34] can compile SBOL specifications into SBML models. Similarly, Virtual Parts Repository uses SBOL specifications to combine existing SBML models together [35]. The rule-based modeling framework BioNetGen [36] allows for a system to be defined via interaction rules which can then be simulated directly or compiled into a CRN. Similarly, PySB [37] provides a library of common biological parts and interactions that compile into more complex rule-based models. Finally, the MATLAB TX-TL Toolbox [38, 39] can be seen as a prototype for BioCRNpyler but lacks the object-oriented framework and extendability beyond cell-free extract systems.

BioCRNpyler compliments existing software packages by providing a novel abstraction and framework which allows for complex CRNs to be easily generated and explored via the compilation process. To do this, BioCRNpyler specifies a biochemical system as a set of modular biological parts, biochemical processes codified as CRNs, and biochemical and modeling context. Moreover, BioCRNpyler allows for synthetic biological parts and systems biology motifs to be reused and recombined in diverse biochemical contexts at customizable levels of model complexity with minimal coding requirements (BioCRNpyler is designed to be a scripting language). Additionally, BioCRNpyler is purposefully suited to *in silico* workflows because it is an extendable object-oriented framework written entirely in Python that integrates existing software development standards and allows complete control over model compilation. Simultaneously, BioCRNpyler accelerates model construction with extensive libraries of biochemical parts, models, and examples relevant to synthetic biologists, bio-engineers, and systems biologists. The BioCRNpyler package is available on GitHub [40] and can be installed via the Python package index (PyPi).

## 2 Design and implementation

BioCRNpyler is an open-source Python package that compiles high-level design specifications into detailed CRN models, which then are saved as SBML files [41]. BioCRNpyler is written in Python with a flexible object-oriented design, extensive documentation, and detailed examples which allow for easy model construction by modelers, customization and extension by

developers, and rapid integration into data pipelines. The utility of BioCRNpyler comes from the way it abstracts biological systems using modular objects. A BioCRNpyler model consists of a collection of biological parts called `Components` which interact via different biological processes called `Mechanisms`. Sets of `Components` and `Mechanisms` are bundled together to form a system, called a `Mixture`, which represents a specific biological and modeling context. During compilation, each `Component` in a `Mixture` generates the species and reactions which model its behavior using `Mechanisms`. This abstraction is powerful; it allows modelers to examine how a specific system, represented by one or more `Components`, behaves in diverse environments and/or under different modeling assumptions represented by different `Mixtures`. Importantly, `Mechanisms` provide a universal underlying abstraction used to define both the way `Components` and `Mixtures` function. In the following subsections, we describe the BioCRNpyler modeling abstraction in detail.

## 2.1 Internal CRN representation

Underlying BioCRNpyler is a comprehensive chemical reaction network class. The species classes in BioCRNpyler consist of object-oriented data structures with increasing complexity which generate their own unique string representations. Table A in S1 Text describes the different species classes in BioCRNpyler. Similarly, BioCRNpyler comes equipped with many diverse propensity function types including mass-action, Hill functions, and general user specified propensities described in Table B in S1 Text. The CRN classes inside BioCRNpyler provide useful functionality so that users can easily modify CRNs produced via compilation, produce entire CRNs by hand, or interface hand-produced CRNs with compiled CRNs. Additionally, user-friendly printing functionality allows for the easy visualization of CRNs in multiple text formats or as interactive reaction graphs formatted and drawn using Bokeh and ForceAtlas2 [42, 43].

## 2.2 Mechanisms are reaction schemas

When modeling biological systems, modelers frequently make use of mass-action CRN kinetics which ensure that parameters and states have clear underlying mechanistic meanings. However, for the design of synthetic biological circuits and analysis using experimental data, phenomenological or reduced-order models are commonly utilized as well [2]. Empirical phenomenological models have been successful in predicting and analyzing complex circuit behavior using simple models with only a few lumped parameters [44–46]. Bridging the connections between the different modeling abstractions is a challenging research problem. This has been explored in the literature using various approaches such as by direct mathematical comparison of mechanistic and phenomenological models [47–49] or by studying particular examples of reduced models [2]. BioCRNpyler provides a computational approach using reaction schemas to easily change the mechanisms used in compilation from detailed mass-action to coarse-grained at various level of complexity.

Reaction schemas refer to BioCRNpyler's generalization of switching between different mechanistic models: a single process can be modeled using multiple underlying motifs to generate a class of models which may have qualitatively different behavior. `Mechanisms` are the BioCRNpyler objects responsible for defining reaction schemas. In other words, various levels of abstractions and model reductions can all be represented easily by using built-in and custom `Mechanisms` in BioCRNpyler. Biologically, reaction schemas can represent different underlying biochemical mechanisms or modeling assumptions and simplifications. For example, to model the process of transcription (as shown in Fig 1), BioCRNpyler allows the use of various phenomenological and mass-action kinetic models by simply changing the choice of reaction

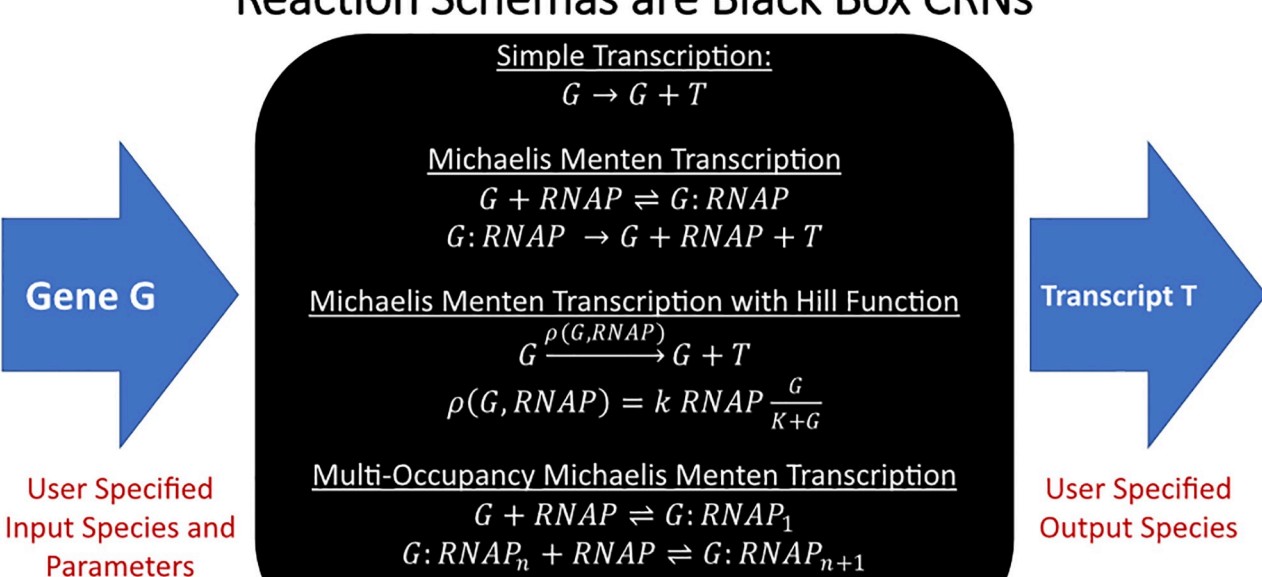

**Fig 1. Mechanisms (reaction schemas) representing transcription.**

schema. The simplest of these schemas "Simple Transcription" includes no details about how a gene produces a transcription. "Michaelis Menten Transcription" elaborates on this simplification by including the RNA polymerase enzyme in the model. "Michaelis Menten Transcription with a Hill Function" simplifies the previous mass action model assuming a quasi-equilibrium approximation of RNA polymerase binding. Finally, the "Multi-Occupancy Michaelis Menten Transcription Model" aims to be more realistic by examining the possibility of multiple RNA polymerase enzymes bound to a single transcript. Of course, these are not the only possible transcription `Mechanisms`: more detailed models may include transcript elongation or organism-specific co-factors, such as $\sigma$-factors in *E. coli*, which could also easily be included in a BioCRNpyler `Mechanism`.

Formally, reaction schemas are functions that produce CRN species and reactions from a set of input species and parameters: $f : (S', \theta) \rightarrow (S, R)$. Here the inputs $S'$ are chemical species and $\theta$ are rate constants. The outputs $S \supseteq S'$ is an increased set of species and $R$ is a set of reactions. The functions $f$ used to define the transcription reaction schemas in Fig 1 are examples of relatively simple `Mechanisms` which do not have any internal logic. However, BioCRNpyler allows for reaction schemas to be defined directly in Python. This allows for incredible flexibility in defining `Mechanisms` capable of complex logic, combinatoric enumeration, or other advanced functionality. The object oriented design of `Mechanisms` also allows modelers to generate CRNs at different levels of complexity and reuse CRN motifs for some `Components` while customizing `Mechanisms` for others. Internally, each `Mechanism` class has a type (e.g. transcription) which defines the input and output species it requires. BioCRNpyler contains an extensive library of `Mechanisms` (Table C in S1 Text) which are easy to repurpose without extensive coding. Custom `Mechanisms` are also easy to define by subclassing `Mechanism` as described in Section I in S1 Text. Ultimately, `Mechanisms` provides a

unique capability to quickly compare system models across various levels of abstraction enabling a more nuanced approach to circuit design and exploring system parameter regimes.

## 2.3 Components represent functionality

In BioCRNpyler, `Components` are biochemical parts or motifs, such as promoters, enzymes and chemical complexes. `Components` represent biomolecular functionality; a promoter enables transcription, enzymes perform catalysis, and chemical complexes must bind together. `Components` express their functionality by calling particular `Mechanism` types during compilation. Importantly, `Components` are not the same as CRN species; one species might be represented by multiple `Components` and a `Component` might produce multiple species. For example, a promoter `Component` will call transcription `Mechanisms` like those shown in Fig 1. If the "Simple Transcription" `Mechanism` is used, the promoter will be represented by a single species *G*. On the other hand, if the "Michaelis Menten Transcription" schema is used, the promoter will actually have two forms: *G* and *G:RNAP* representing the free promoter and the promoter bound to RNA polymerase. `Components` are flexible and can behave differently in different contexts or behave context-independently. To define dynamic-context behavior, `Components` will automatically use `Mechanisms` and parameters provided by the `Mixture`. To define context-independent behavior, `Components` can have their own internal `Mechanisms` and parameters. The BioCRNpyler library includes many kinds of `Component` some of which are listed in S1 Text Table D. Custom `Components` can also be easily created by subclassing another `Component` as described in Section II in S1 Text.

## 2.4 Mixtures represent context

`Mixtures` are collections of `Components`, `Mechanisms`, and parameters. `Mixtures` can represent chemical context (e.g. cell extract vs. *in vivo*), as well as modeling resolution (e.g. what level of detail to model transcription or translation at) by containing different internal `Components`, `Mechanisms`, and parameters. BioCRNpyler comes with a variety of `Mixtures` (see Table E in S1 Text) to represent cell-extracts and cell-like systems with multiple levels of modeling complexity. Custom `Mixtures` can also be easily created either by subclassing an existing mixture or via a few simple scripting operations as described in Section III in S1 Text.

## 2.5 Flexible parameter databases

Developing models is a process that involves defining then parameterizing interactions. Often, at the early stage of model construction, exact parameter values will be unavailable. BioCRNpyler has a sophisticated parameter framework which allows for the software to search user-populated parameter databases for the parameter that closest matches a specific `Mechanism`, `Component`, and parameter name as illustrated in Fig 2. This allows for models to be rapidly constructed and simulated with "ball-park" parameters and then later refined with specific parameters derived from literature or experiments later. This framework also makes it easy to incorporate diverse parameter files together and share parameters between many chemical reactions. BioCRNpyler also allows each `Component` to have its own parameter database allowing for multiple parameter sources to be combined easily. Components without their own parameters default to the parameters stored in the `Mixture`.

# BioCRNpyler Parameter Hierarchy

**Fig 2. BioCRNpyler parameter defaulting hierarchy.** If a specific `ParameterKey` (orange boxes) cannot be found, the `ParameterDatabase` automatically defaults to other `ParameterKeys`. This allows for parameter sharing and rapid construction of complex models from relatively few non-specific (e.g. lower in the hierarchy) parameters.

## 2.6 Component enumeration allows for arbitrary complexity

Component enumeration is a powerful and specialized compilation step which allows new `Components` to be generated dynamically. Internally, this is achieved in BioCRNpyler by subclassing the `ComponentEnumerator` class to implement an arbitrary function in Python $g: C \to C'$ where $C \subset C'$ are sets of `Components`. In local component enumeration the set $C$ consist of just a single component $c$ which contains its own `ComponentEnumerator`. In global component enumeration, $C$ consists of all components in the `Mixture`. As more `Components` are generated, $C'$ will be fed back into $g$ recursively until no new `Components` are created or a user defined recursion depth is reached. Like `Mechanisms`, we emphasize that component enumeration is highly flexible because the enumerators can be written as Python code, allowing for diverse logic, combinatoric enumeration, and more. Section 3.3 describes BioCRNpyler models that makes use of both local and global component enumeration.

## 2.7 Specification example

Before describing the compilation algorithm in detail, we illustrate the central idea of a BioCRNpyler specification via an example involving a `DNAassembly Component` which represents a simple piece of DNA, called *X*, with a promoter, ribosome binding site, and coding sequence for a protein. The `DNAassembly` uses transcription and translation `Mechanisms` which will be placed into a `Mixture`.

```
# Create Mechanisms
tx = SimpleTranscription() #Transcription
```

```
tl = SimpleTranslation() #Translation
# Create a Component
G = DNAassembly("X", promoter = "prom", rbs = "rbs", protein = "X")
# Define Parameters
params = {"kb":100, "ku":10, "ktx":0.1, "ktl":0.5,}
# Place the Component and Mechanisms in a Mixture
M = Mixture("mixture", components = [G], mechanisms = [tx, tl],
            parameters = params)
# Compile the CRN
CRN = M.compile_crn()
```

This simple code compiles the CRN:

$$X_{\text{DNA}} \xrightarrow{0.1} X_{\text{DNA}} + X_{\text{RNA}} \qquad X_{\text{RNA}} \xrightarrow{0.5} X_{\text{RNA}} + X_{\text{Protein}}. \tag{1}$$

The modularity of BioCRNpyler can be illustrated by considering what would happen if we instead used "Michaelis Menten" transcription and translation `Mechanisms` which model RNA-polymerase (*P*) and ribosomes (*R*):

```
tx = Transcription_MM(rnap = Species("P")) #Transcription Mechanism
tl = Translation_MM(ribosome = Species("R")) #Translation Mechanism
```

This compiles a considerably more complex CRN:

$$X_{\text{DNA}} + P \underset{10}{\overset{100}{\rightleftharpoons}} X_{\text{DNA}} : P \xrightarrow{0.1} X_{\text{DNA}} + P + X_{\text{RNA}}$$

$$X_{\text{RNA}} + R \underset{10}{\overset{100}{\rightleftharpoons}} X_{\text{RNA}} : R \xrightarrow{0.5} X_{\text{RNA}} + R + X_{\text{Protein}}.$$

Here, ":" indicates that two species are bound together to form a new species.

## 2.8 Chemical reaction network compilation

Having provided an overview of the core classes in BioCRNpyler, we will now describe the compilation algorithm in detail. First, we assume a user has specified a `Mixture` and populated it with `Components`, `Mechanisms`, and parameters. We note that some `Components` may have their own internal `Mechanisms` and `Parameters` while others will be reliant on the `Mixture`. Compilation proceeds in 7 steps, shown in Fig 3 and elaborated on below.

1. **Global Component Enumeration**: this step is optional and will only occur if a `Mixture` contains a one or more global `ComponentEnumerators`. All `Components` in the `Mixture` will be fed into the `ComponentEnumerator` recursively until either no new `Components` are created or a user-specified recursion depth is reached.

2. **Local Component Enumeration**: this step is optional and will be applied to every `Component` in the `Mixture` that contains a one or more local `ComponentEnumerators`. Each of these `Components` will generate new `Components` from itself. If these new `Components` contain local `ComponentEnumerators` they will also generate new `Components`. Like global component enumeration, local component enumeration is stopped when no new `Components` are created or a user-specified maximum recursion depth is reached.

3. The `Mixture` iterates through all its internal `Components` (including those generated via enumeration) and calls the `Component`'s `update_species()` and `update_reactions()` methods.

4. In each `Component`'s `update_species()` and `update_reactions()` method, the `Component` first searches for `Mechanisms` of the types it requires. `Mechanisms`

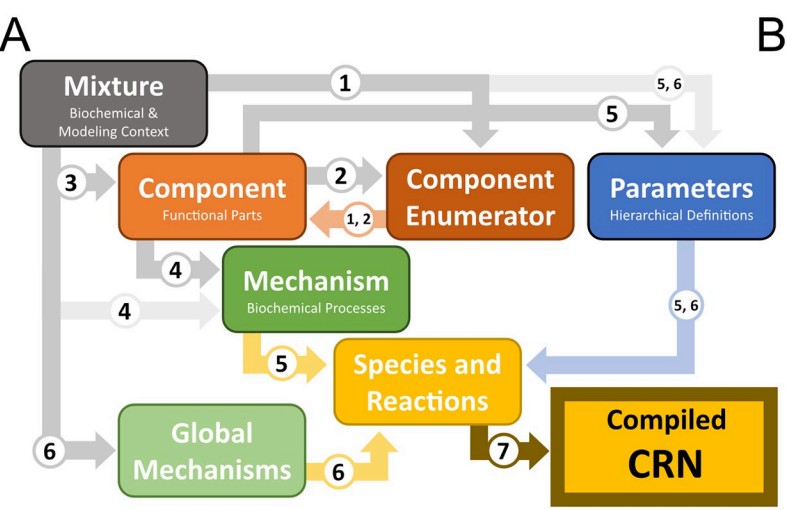

**Fig 3. A**. the organization of classes in BioCRNpyler. Gray arrows indicate the hierarchical organization of objects (e.g. `Components` are contained in a `Mixture`). Dark gray arrows take precedence over light gray arrows (e.g. a `Component` will search for `Mechanisms` in itself before looking at its `Mixture`). Colored arrows denote the generate of objects: `Components` are orange, parameters are blue, and CRN species and reactions are yellow. **B**. The compilation sequence in BioCRNpyler. The numbers on the arrows in (A) indicate which part of compilation these connections are involved in.

stored inside the `Component` will be used preferentially. If the `Component` does not have a particular internal `Mechanism`, that `Mechanism` is instead retrieved from the `Mixture`. The `Component` then calls the `update_species(...)` and `update_reactions(...)` methods of each `Mechanism` supplying the proper parameters for that `Mechanism`.

5. `Mechanisms` generate species and reactions based upon the arguments supplied by the `Component` that called them. `Mechanisms` search for rate parameters in the parameter database of the `Component` that called them. If no parameters are found, the `Mechanism` will then search for parameters in the `Mixture`'s parameter database. Note that the same `Mechanism` may be called multiple times with different parameters, effectively reusing the reaction schema to compile a large CRN. The species and reactions generated this way are returned to the `Mixture`.

6. Global `Mechanisms` are a special kind of `Mechanism` which are stored in the `Mixture` and produce new species and reactions from a single species parameter. All species generated in previous steps are passed into the `Mixture`'s global `Mechanisms` to generate additional species and reactions. Note that global `Mechanisms` are not called recursively.

7. The resulting species and reactions generated in the previous steps form a chemical reaction network which can be modified programatically or exported as SBML.

## 2.9 Integrated testing

BioCRNpyler uses GitHub Actions and Codecov [50] to automate testing on GitHub. Whenever the software is updated, a suite of tests is run including extensive unit tests and functional testing of tutorial and documentation notebooks. Automated testing ensures that changes to the core BioCRNpyler code preserve functionality of the package. The integration of Jupyter notebooks into testing allows users to easily define new functionality for the software and document that functionality with detailed explanations which are simultaneously tests cases.

## 2.10 Documentation and tutorials

The BioCRNpyler GitHub page contains over a dozen tutorial Jupyter notebooks [40] and video presentations explaining everything from the fundamental features of the code to specialized functionality for advanced models to how to add to the BioCRNpyler code-base [51]. This documentation has been used successfully in multiple academic courses and is guaranteed to be up-to-date and functional due to automatic testing.

# 3 Results

This section highlights the functionality of BioCRNpyler through a collection of models compiled using the software. All model simulations were conducted with Bioscrape [52], circuit diagrams were created with DNAplotlib [53], and reaction network graphs were created with BioCRNpyler's plotting interface. Detailed descriptions alongside commented code for all the following examples are available in S1 Text Section A and as Jupyter notebooks on the BioCRNpyler GitHub page.

## 3.1 Synthetic biological circuit examples

Fig 4A, 4B and 4C show three models of synthetic biological circuits which demonstrate the modularity and expressivity of BioCRNpyler. Underlying all these models is a single `Component` class called a `DNAassembly` which was described in Section 2.7. These first three examples use idealized models of their underlying biological processes via a very simple `Mixture`. In Fig 4A two `DNAassemblies` are wired together with a repressor (red) repressing a report (yellow). The repressor is expressed at a constant rate using the "Simple Transcription" `Mechanism` shown in Fig 1 which is supplied by the `Mixture`. The reporter, on the other hand, uses a different transcription `Mechanism`, "Negative Hill Repression" stored in its `DNAassembly`. This illustrates the ability for the same process, transcription, to be modeled in different ways within a single model. In Fig 4B, two `DNAassembly Components` are wired to repress each other, both using Hill functions, to produce a model of the famous bistable toggle switch [54]. Similarly, Fig 4C wires three repressors together so A represses B, B represses C, and C represses A, giving rise to a transcriptional oscillator called the repressilator [55].

Fig 4D, 4E and 4F examine similar circuits to Fig 4A, 4B and 4C but with more complex implementations modeled in a more detailed context. In these three following examples, a less idealized `Mixture` is used which models transcription, translation, and RNA degradation with biological machinery including RNA polymerase, ribosomes, and RNAses. Fig 4D examines a detailed implementation of a repression circuit consisting `DNAassembly Components` which express a guide-RNA (gRNA) and deactived Cas9 (dCas9) protein [56]. The dCas9-gRNA complex is capable of binding to the promoter of the reporter assembly, repressing transcription. This more complex circuit in a complex context reveals some unexpected behavior; if the amount of dCas9 and gRNA are not carefully balanced, resource loading can give rise to unexpected increases and decreases of the reporter, a phenomena known as retroactivity [57]. Fig 4E shows a hypothetical variation of a bistable toggle switch implemented via translational regulation using targeted RNAses (RNAse A degrades the transcript for RNAse B and visa-versa). Such a system could potentially be engineered via RNA-targeting Cas9 [58] or more complex fusion proteins [59]. Finally, Fig 4F compiles a model of the repressilator which allows for multiple ribosomes to bind to each transcript. The added complexity creates much more complicated dynamics, but oscillatory behavior still clearly occurs. This example illustrates how BioCRNpyler can be used to test different modeling assumptions (e.g. does multiple occupancy of ribosomes matter?).

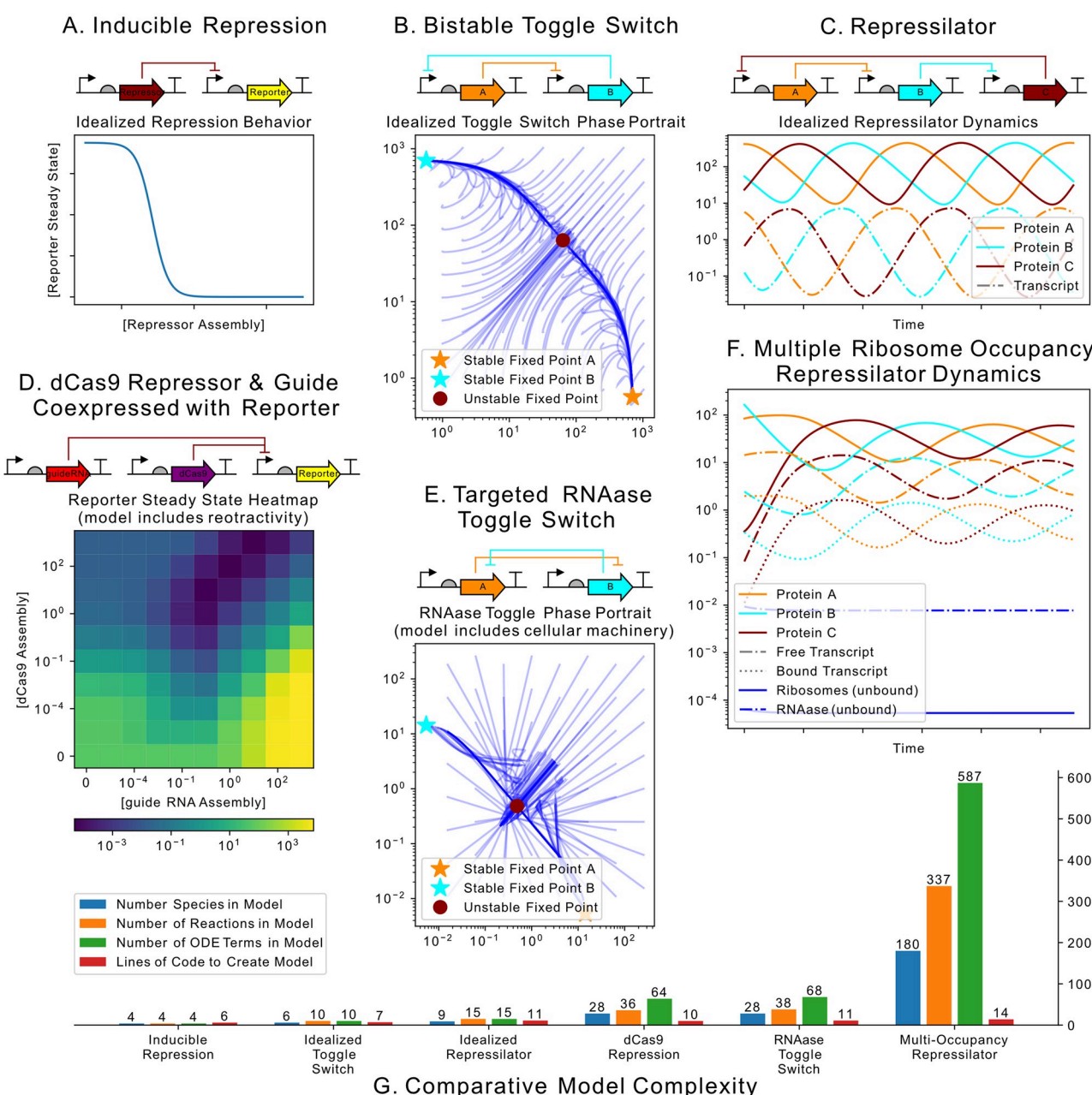

**Fig 4. Motivating examples.** The idealized models (A, B, and C) do not model the cellular environment; genes and transcripts transcribe and translate catalytically. **A**. Schematic and simulation of a constituitively active repressor gene repressing a reporter. **B**. Schematic and simulations of of a toggle switch created by having two genes, *A* and *B*, mutually repress each other. **C**. Schematic and dynamics of a 3-repressor oscillator. The detailed models (D, E, & F) model the cellular environment by including ribosomes, RNAases and background resource competition for cellular resources. **D**. A dCas9-guideRNA complex binds to the promoter of a reporter and inhibiting transcription. Heatmap shows retroactivity caused by varying the amount of dCas9 and guide-RNA expressed. The sharing of transcription and translational resources gives rise to increases and decreases of reporter even when there is very little repressor. **E**. A proposed model for a non-transcriptional toggle switch formed by homodimer-RNAase; the homodimer-RNAase made from subunit *A* selectively degrades the mRNA producing subunit *B* and visa-versa. **F**. A model of the Repressillator exploring the effects of multiple ribosomes binding to the same mRNA. **G**. Histogram comparing the sizes of models A-F and the amount of BioCRNpyler code needed to generate them.

Finally, we comment that all the examples from Fig 4 make use of the same underlying set of 10–20 default parameters (estimated from Cell Biology by the Numbers [60]) demonstrating how BioCRNpyler's parameter database and defaulting behavior make model construction and simulation possible even before detailed experiments or literature review. The efficiency of using BioCRNpyler to explore diverse modeling assumptions and circuit architectures is quantified in Fig 4G which compares the number of species, reactions, and ordinary differential equation terms in the compiled models to the lines of BioCRNpyler code needed to create these models. In short, BioCRNpyler allows for the rapid generation of large and diverse models. Code for these six examples can be in Sections I-IV in S1 Text.

### 3.2 Systems biology circuit example

Fig 5 illustrates how a set of BioCRNpyler `Components` and `Mechanisms` can be joined together to produce a systems level model of the lac operon—a highly studied gene regulatory network in *E. coli* which regulates whether glucose or lactose is metabolized [61]. This specification is shown in Fig 5A and consists of around a dozen `Components` and `Mechanisms`

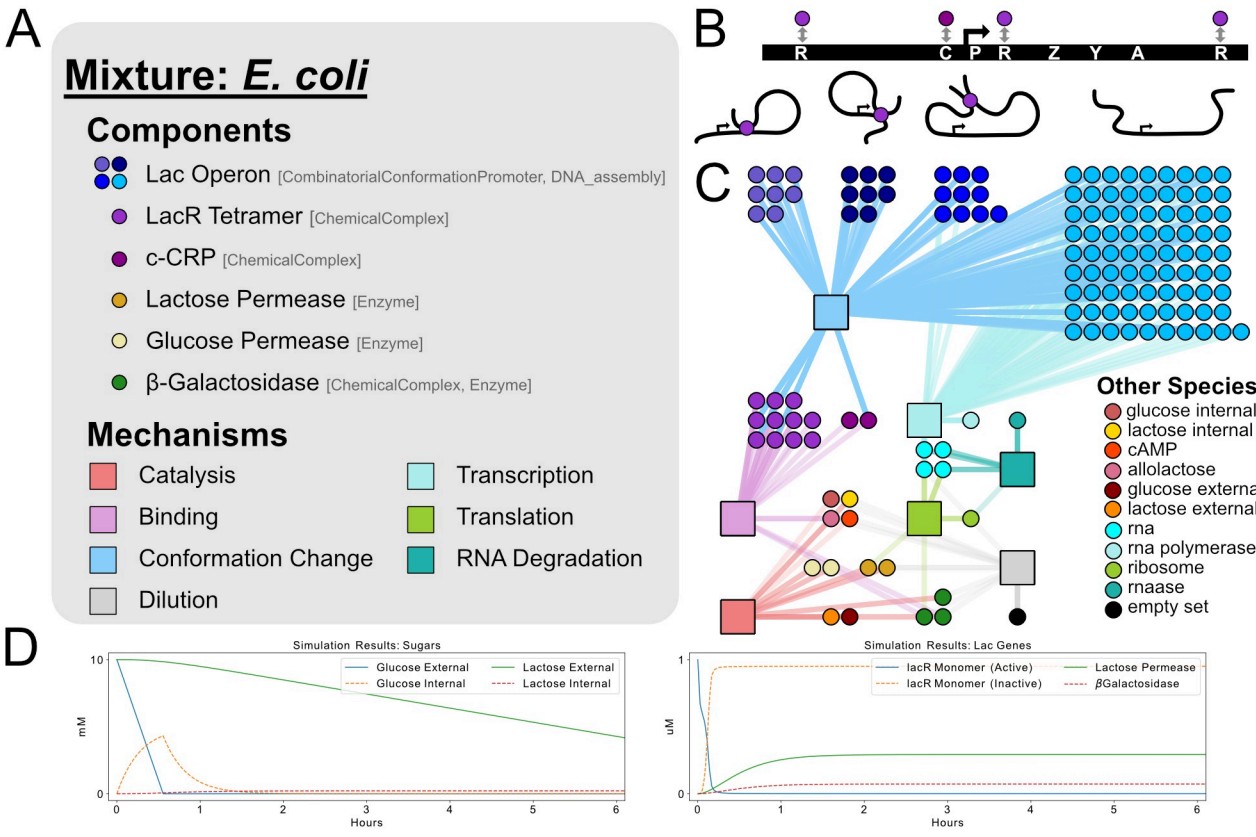

**Fig 5. A model of the lac operon compiled using BioCRNpyler specifications with 141 species and 271 reactions using ∼50 lines of code. A**. A `Mixture` contains a set of `Components` and `Mechanisms`. The `Component` classes used for each element of the model are shown in brackets. The colored circles show how `Components` correspond to compiled CRN species in panel C. **B**. A schematic of the lac operon and the three looped and one open conformation it can take. Each conformation contains a combinatoric number of states based upon the accessible binding sites: R are lac repressor binding sites; C is the activator c-CRP binding site; P is the promoter; and Z, Y, A are the three lac genes. The conformations are placed over clusters of identically colored species corresponding to that conformation in the compiled CRN. **C**. A graph representation of the compiled CRN. Each circle is a unique chemical species. Square boxes show how chemical species interact via reactions generated by specific `Mechanisms`. **D**. Simulated output of the model.

which jointly enumerate hundreds of species and reactions representing the combinatorial set of conformations of the lac operon (depicted by the cartoons in panel B) and its associated transcription factors, transcription, translation, transport, mRNA degradation and dilution. Besides showing how BioCRNpyler can be applied to model the kinds of combinatoric interactions common in systems biology, this example also graphically illustrates the BioCRNpyler abstraction where `Components` interact via `Mechanisms` in order to generate a large, complex CRN (panel C). Furthermore, this example highlights that the `Component` species mapping is not one-to-one. For example, the Lac Operon is modeled as two `Components` one representing the promoter architecture and another coupling that promoter to translation. Jointly, these two `Components` produce a combinatoric number of formal CRN species (shown in panel C by the many different blue dots). Similarly, $\beta$-galactosidase is modeled as two `Components`: as an enzyme (which metabolizes lactose) and a chemical complex (because it is a homeotetramer). Finally, we note that the simulated output of our model (Fig 5D) produces a $\sim$1-2 hour delay between the depletion of glucose and steady state lactose metabolism, consistent with previous models and experiments [61]. Interestingly, this is observed even though we made no efforts to fine-tune our parameters, suggesting that the combinatorial nature of this system may give rise to this behavior in a manner that is robust to detailed kinetic rates. The code used to generate this model can be found in Section VII in S1 Text.

### 3.3 Component enumeration example

Fig 6 shows three example circuits which make use of component enumeration in order to produce sophisticated CRNs. Local component enumeration is illustrated in Fig 6A. Here, a single DNA `Component` (top) uses local component enumeration to read through the parts included in its plasmid and determine all possible correctly oriented terminator-promoter pairs. This information is then used to produce multiple RNA `Components` which model transcription and translation for complex genetic circuit architectures. The CRN and simulation output for this circuit are shown in Fig 6B and 6C, respectively. Fig 6D provides an example of global component enumeration involving the enzymatic recombination of DNA. Specifically, serine integrases (such as Bxb1) are enzymes capable of recombining strands of DNA at specific integration sites [62]. Integration events can happen within a single piece of DNA (top two reactions in panel D) or between multiple DNA species (bottom 4 reactions of panel D). In these reactions, the integrase binds to attP and attB sites and reorganizes them into attL and attR sites which can result in DNA insertions, excisions, or re-orientations. Importantly, each new DNA strand produced by an integrase reaction could potentially recombine with itself or the other strands already produced. Such systems can give rise to theoretically infinite CRNs [63]. BioCRNpyler can approximate integrase systems by recursively using a global component enumerator. In this example, only a single round of recursion is shown for clarity. The clusters of dots in Fig 6E are due to the combinatoric number bound and unbound states due to the potential for integrases to bind and unbind to attP, attB, attL, and attR sites. Finally, the BioCRNpyler framework is designed so that local and global component enumeration are mutually compatible. In Fig 6F, a model of a self-flipping promoter is shown. Initially, the promoter faces right and expresses the integrase Bxb1 which in turn flips the promoter causing Bxb1 expression to cease in favor of RFP expression. In BioCRNpyler, this model is compiled by first using global component enumeration to produce all the possible DNA `Components` generated by integrase recombinations. Each of these DNA `Components` then uses local component enumeration to produce RNA `Components`. All these `Components` can then be used to compile a CRN by calling their respective `Mechanisms`.

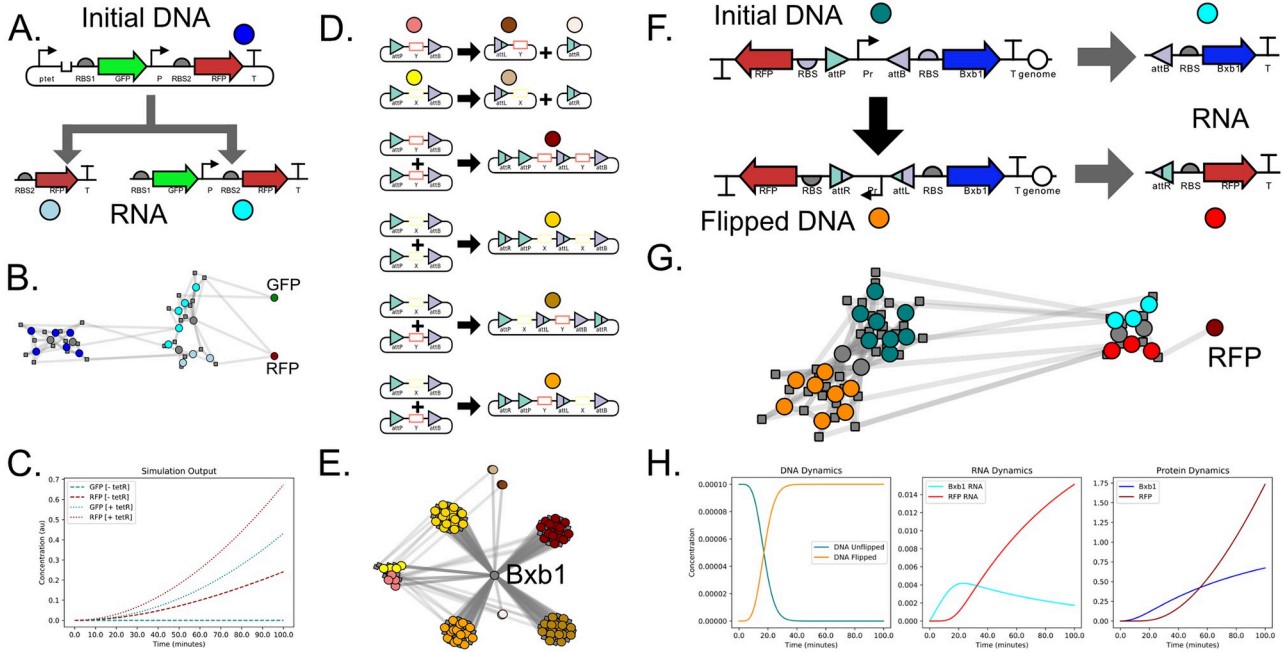

**Fig 6. Examples involving component enumeration. A**. Schematic of local component enumeration for a gene expression circuit where a single DNA `Component` generates multiple RNA `Components`. **B**. The CRN for (A) represented graphically. Colored dots are species corresponding to the components adjacent to the dots in (A). **C**. Simulated output from the CRN in (B). **D**. Schematic of global component enumeration in an integrase circuit where one or more DNA `Components` recombine to produce new DNA `Components`. Note that the larger DNA outputs could also recombined analogously but this is not shown. **E**. The CRN for (D) represented graphically. Colored dots are species which correspond to the components adjacent to the dots in (D). **F**. A genetic circuit which combines global and local component enumeration to flip a promoter which drives gene expression. **G**. The CRN for the circuit in (F). Colored dots are species representing the components adjacent to the dots in (F). **H**. Simulated output of the CRN.

More details about local and global component enumeration, including code for the example models, can be found in Sections VIII-X in S1 Text.

## 4 Availability and future directions

BioCRNpyler aims to be a piece of open-source community driven software that is easily accessible to biologists and bioengineers with varying levels of programming experience as well as easily customizable by computational biologists and more advanced developers. Towards these ends, the software package is available via GitHub and PyPi, requires very minimal software dependencies, contains extensive examples and documentation in the form of interactive Jupyter notebooks [40], YouTube tutorials [51], and automated testing to ensure stability. Furthermore this software has been extensively tested via inclusion in bio-modeling courses and bootcamps for users ranging from college freshmen and sophomores with minimal coding experience to advanced computational biologists demonstrating the accessibility and flexibility of the package. BioCRNpyler has already been deployed to build diverse models in synthetic biology including modeling bacterial gene regulatory networks [64], modeling bacterial circuits in the gut microbiome [65], and modeling cell extract metabolism [66]. Developing new software functionality is also a simple process documented on the GitHub contributions page.

**Table 1. Comparison of different simulation software. Abstraction**: how models can be represented in the software. **Library**: whether there is a substantial library of pre-existing parts/components/sub-models that can be reused. **Simulator**: whether the software simulates models numerically. **Source**: the language(s) the software is written in. **UI**: the primary way a user interacts with the software. **API**: the primary programming language the software is designed to be accessed with.

| Software | Abstraction | Library | Simulator | Source | UI | API |
|---|---|---|---|---|---|---|
| **BioCRNpyler** [40] | `Mixtures`, `Components`, `Mechanisms`, & CRNs | Yes | No | Python | Python | Python |
| **BioNetGen** [36] | Rules | No | Yes | Perl C+ + Python | .bng files | .bng files |
| **PySB** [37] | Rules | Yes | No | Python | Text Rules | Python |
| **Tellurium** [12] (using Antimony [30] and libRoadrunner [14]) | CRNs | No | Yes | Python | Text Reactions | Python |
| **Virtual Parts Repository** [35] | SBOL | Yes | No | Java | Web | Java |
| **iBioSim** [34] | SBOL & CRNs | Yes | Yes | Java | GUI | Command line |
| **COPASI** [11] | CRNs | No | Yes | Java C++ | GUI | C++ & other derived APIs |
| **MATLAB Simbiology** [13] | CRNs | No | Yes | MATLAB | MATLAB | MATLAB |

Given the plethora of model building and simulation software already in existence, it is important to highlight how BioCRNpyler fits into the larger context of existing tools. Table 1 gives a high level overview of how BioCRNpyler compares to other tools. Firstly, BioCRNpyler stands out due to the novel `Mixture-Component-Mechanism` abstraction. This framework allows users to easily put together complex models using BioCRNpyler's extensive library or to develop their own extensions by writing Python code. Rule based frameworks, such as BioNetGen [36] and PySB [37] offer similar abstractions to `Mechanisms`. However, these must be codified in a formal language specific to the framework (BioNetGen uses .bng files and PySB uses a specialized text format) which offers less flexibility than the arbitrary python code allowed by BioCRNpyler. The Virtual Parts Repository [35] and iBioSim [34] take a different approach to abstract specifications by generating CRNs from SBOL files. This methodology is similar in spirit to BioCRNpyler but is restricted due to the reliance on the SBOL standard, the need of software-specific SBOL annotations, and challenges in generalizing beyond gene regulatory network architectures. BioCRNpyler also differs from many other pieces of software because it includes a detailed library of biological parts and models. PySB, Virtual Parts Repository, and iBioSim similarly include a variety of built-in rules, models, and parts, respectively. However, BioCRNpyler is unique in its modularity: the ability to use the same `Component` with different `Mechanisms` placed in different `Mixtures` allows for a combinatoric variety of models to be easily specified and explored. Finally, we reiterate that BioCRNpyler is not a CRN simulator like COPASI [11], MATLAB Simbiology [13], or Tellurium (via libroadrunner) [12, 14]. This brings us to a final point about BioCRNpyler: it is a pure Python package with very minimal dependencies meant to be used as a scripting language, interfaced with existing simulators, used in Jupyter notebooks [67], and integrated into existing pipelines.

BioCRNpyler is an ongoing effort which will grow and change with the needs of its community. Extending this community via outreach, documentation, and an ever expanding suite of functionalities is central to the goals of this project. We are particularly interested in facilitating the integration of BioCRNpyler into existing laboratory pipelines in order to make modeling a central part of the design-build-test cycle in synthetic biology. One avenue towards this goal is to add compatibility to existing standards such as SBOL [31] and automation platforms such as DNA-BOT [68] so BioCRNpyler can automatically compile models of circuits as they are being designed and built. This approach will be a generalization and extension of Roehner

et al. [69]. In particular due to the modular BioCRNpyler compilation process, it will be possible to have programmatic control over the SBML model produced from BioCRNpyler.

We also plan on extending the library to include more realistic and diverse `Mixtures`, `Mechanisms`, and `Components` (particularly experimentally validated models of circuits in *E. coli* and in cell extracts). We hope that these models will serve as examples and inspiration for other scientists to add their own model systems in other organisms to the software library.

Finally, we believe that the `Mixture-Component-Mechanism` abstraction of model compilation used in BioCRNpyler is quite fundamental and could be extended to other non-CRN based modeling approaches. Advanced simulation techniques beyond chemical reaction networks will be required to accurately model the diversity and complexity of biological systems. New software frameworks such as Vivarium [64] have the potential to generate models which couple many simulation modalities. The abstractions used in BioCRNpyler could be extended to compile models beyond chemical reaction networks such as mechanical models, flux balance models, and statistical models derived from data. The integration of these models together will naturally depend on both detailed mechanistic descriptions as well as overarching system context. We emphasize that building extendable and reusable frameworks to enable quantitative modeling in biology will become increasingly necessary to understand and design ever more complex biochemical systems.

## Supporting information

**S1 Text.** Table A: (CRN Species Classes the BioCRNpyler Library). Table B: (Reaction Propensities in the BioCRNpyler Library). Table C: (Some Mechanisms in the BioCRNpyler Library). Table D: (Some Components in the BioCRNpyler Library). Table E: (Some Mixtures in the BioCRNpyler. Library).
(PDF)

## Acknowledgments

We would like to thank the https://murray.cds.caltech.edu/BE_240,_Spring_2020 and the Murray Biocircuits lab for extensive testing of this software and discussions of relevant models, library of parts, and parameters. In particular, we would like to thank Zoila Jurado, Matthieu Kratz, Liana Merk, and Ankita Roychoudhury for contributing to the software library.

## Author Contributions

**Conceptualization:** William Poole, Ayush Pandey, Richard M. Murray.

**Funding acquisition:** Richard M. Murray.

**Investigation:** William Poole.

**Methodology:** William Poole, Ayush Pandey, Andrey Shur, Zoltan A. Tuza, Richard M. Murray.

**Project administration:** William Poole, Richard M. Murray.

**Software:** William Poole, Ayush Pandey, Andrey Shur, Zoltan A. Tuza.

**Visualization:** William Poole, Andrey Shur.

**Writing – original draft:** William Poole.

**Writing – review & editing:** William Poole, Ayush Pandey, Andrey Shur, Zoltan A. Tuza, Richard M. Murray.

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
