## [Decision Letter · Decision Letter 0]

4 Oct 2021

Dear Mr. Poole,

Thank you very much for submitting your manuscript "BioCRNpyler: Compiling Chemical Reaction Networks from Biomolecular Parts in Diverse Contexts" for consideration at PLOS Computational Biology.

As with all papers reviewed by the journal, your manuscript was reviewed by members of the editorial board and by several independent reviewers. In light of the reviews (below this email), we would like to invite the resubmission of a significantly-revised version that takes into account the reviewers' comments.

Please follow reviewer's suggestions regarding the manuscript organization: the manuscript should explain the functionality of the BioCRNpyler, while the manual and tutorial should be part of the github repository.

We cannot make any decision about publication until we have seen the revised manuscript and your response to the reviewers' comments. Your revised manuscript is also likely to be sent to reviewers for further evaluation.

Sincerely,

Pedro Mendes, PhD

Associate Editor

PLOS Computational Biology

Dina Schneidman

Software Editor

PLOS Computational Biology

Reviewer's Responses to Questions

**Comments to the Authors:**

Reviewer #1: The paper presents a new compiler from genetic circuit designs specified in a high-level language into chemical reaction network models appropriate for simulation and other analyses. This high-level language and compiler can greatly reduce the size of the description of practical genetic circuit designs as demonstrated in their results shown in Figure 1G. Therefore, this tool is likely to prove to be very useful. The authors are commended for providing this open source with detailed documentation, examples, and tutorials.

The paper though needs some work to better articulate their contributions while positioning themselves with respect to the related work. Here are my more detailed comments:

1) Abstract (and introduction) - you talk about the high-level design specification, which is your key contribution, but very little detail about it is provided in the abstract or introduction. It would be useful to provide a bit of a description in both the abstract and introduction, so the reader can understand at what level of abstraction it works and provide some intuition why it may result in reductions in model complexity. A small motivating example early in the paper would likely be useful to engage the reader sooner. Indeed, all the examples of the actual specification seem to only appear in the supplemental.

2) Intro - "relatively few tools exist to aid in the automated construction of general CRN models from simple specifications", I'm not so sure I completely agree with this statement. There are several tools that the authors do not reference that do just this, such as Antimony, ShortBOL, and the Virtual Parts Repository. Even pySBOL coupled with model generation tools such as VPR or iBioSim provide similar functionality to what they are proposing. I do believe that their approach is novel and useful, but the authors need to do a better job articulating the similarities and differences with these other approaches.

3) Page 8 and a few other places, "objected-oriented" -> "object-oriented"

4) Figure 1G is the key result demonstrating the utility of this work. A bit more intuition for this result is needed. Part of the issue is not having a detailed example up to this point.

5) Figure 6, repeated "relatively" in the caption.

6) A key contribution cited is the flexibility to have alternative models. My understanding is this requires writing python code to create new mechanisms. In some sense, this lessens the impact of this contribution, since other model generators can (and do) provide alternative modeling methods via added code to their model generators. It is unclear to me how much easier it is to develop a new model using their approach versus other model generators. A detailed example of how this is done would be useful to demonstrate this.

Overall, this is a useful tool that may have the potential to enable model-driven design for biologists and bioengineers with limited programming experience. The authors need to be clear that some experience is still needed though, as python programs still need to be created. The authors need to also better articulate the differences between their tool and similar approaches.

Reviewer #2: The manuscript describes a Python package for creating of reaction networks in SBML from a high-level design specification. It will be very useful for biologists using Python for modeling.

Pros:

- Easy to install, no extra libraries were required on Mac. A lot of tutorials and examples.

- SBML is correct, tested by importing into COPASI general simulator. Simulations run nicely with LibRoadRunner from Jupyter notebook.

- The libraries of components and mechanisms are useful for compiling multiple models from standard parts.

- Creating new components and mechanisms is very useful way to extend these libraries.

- A very interesting and useful approach of maintaining a parameter database with hierarchy of components and mechanisms (types have multiple names) and automatic substitution of parameters using this hierarchy, for example if no parameter exists for a specific mechanism name, but a parameter value exists for a mechanism type.

Major issues:

1. The manuscript is rather difficult to read. It’s neither a biological paper describing a use of a modeling approach, nor a computational paper describing a software.

a. The manuscript lists a lot of Python classes but does not give any details on how they work. The only way to understand how the package works is to follow examples on GitHub and run them one by one. For example, the authors mention OrderedPolymerSpecies and a PolymerConformation, but never explain what are these, or how DNAconstruct enumerates parts. Most classes are defined so briefly that it’s impossible to understand what they do. “GlobalMechanisms are rules used to generate Species and Reactions at the end of compilation” – it is repeated three times in different contexts, but it does not help with understanding of how it works. …DillutionMixture is neither defined biologically nor explained programmatically, despite being used many times in different contexts. I just recommend the authors to look for any complicated Python class name in the manuscript and ask themselves whether the description in the manuscript is enough to understand the term.

b. The use of the term “hierarchical” and mentioning SBML hierarchical extension are misleading: BioCRNpyler is not generating hierarchical models.

c. There are many GitHub folders with many tutorials. Some guide on which classes are described in which tutorials would be helpful.

2. The authors mention several tools that are comparable to BioCRNpyler, but don’t compare and don’t demonstrate current (not potential) advantages over other tools.

a. At which point the use of BioCRNpyler becomes easier than specifying models in SBML simulators like Copasi and Tellurium? The complex model of Lac Operon is definitely easier to define in any rule-based language, but is BioCRNpyler better than BioNetGen or PySB?

b. What’s the difference with PySB? I noticed parameters, but otherwise the same mechanisms can be defined in PySB, and using BioNetGen in PySB is more powerful.

c. What rule-based features are used? Is it a plain combination of all components without any constraints?

d. More comparing with BioNetGen would be useful – are there any advantages of BioCRNpyler high-level specification over the BNGL language? Can it specify something that BioNetGen cannot?

e. Comparing with iBioSim would be helpful. It also can define genetic components and mechanisms.

f. Comparing with Tellurium/Antimony would be helpful – it has human-readable language.

g. The manuscript will gain a lot if an example will be provided (may be as a supplement) in all four languages: BioCRNpyler, PySB, BioNetGen, and Antimony.

3. The only biological use of BioCRNpyler is in Ref 56, but it is not discussed in the manuscript, all examples are just test examples that repeat well-known and many times modelled biological systems that are simple to be defined in any biomodelling simulator.

4. Lac Operon model is the most complicated model described, but the authors don’t mention what to do with their model of 173 species and 343 reactions, is it comparable with any previous models? And then, I could not find the code for this specific model among examples.

Minor issues:

1. Biorxiv 50, 55-57 – provide full citations.

2. Ref 54 is just a review mentioning BioCRNpyler, it does not demonstrate that “BioCRNpyler has already been deployed to build diverse models in systems and synthetic biology.

3. Ref 55 is not using BioCRNpyler.

4. Why the new term “mixture” is introduced instead of a classical “model”?

Reviewer #3: This work by Poole et al. introduces BioCRNpyler as a tool for users to build reaction networks from high-level design specifications. The tool seems to automate several steps of the network-building process and provides a library of biochemical reactions to build networks using these components. The user is also offered a library of parameters which provides a good starting point for simulation.

Overall, I really wanted to like this work but I feel the authors missed a chance to present their work and build enthusiasm for their tool. The paper itself is structured in a way that makes it hard to understand what and how the tool works and what the benefit of the tool would be for a reader. Rather the paper reads like a user manual with many examples/tutorials rather than a narrative about the tool.

Major comments:

1. The introduction provides a good context for how CRNs are used but it does not provide a compelling argument for why BioCRNpyler is needed. Why is compiling reactions better than what other tools do? Why would the end-user pick BioCRNpyler over other tools? I think a brief introduction to molecular compilers and their use in DNA circuits could help place the tool in context for the reader.

2. In the introduction, the authors mention what existing tools are comparable to BioCRNpyler. I think that a table in the results section would be better suited to provide this information, along with some benchmarks in the supplement.

3. The authors provide a "laundry list" of motivating examples to describe BioCRNpyler. However, these examples are hard to follow. First, the biology context is not very familiar for most readers. Second, Each section is one or two paragraphs with a large amount of data/figures, making it hard to follow for readers. Third, the authors reference software calls without context. This all makes it very hard to follow.

4. Figure 2 is not very informative. It is meant to provide a hierarchical organization of BioCRNpyler but it left me feeling lost. What am I supposed to learn from this figure? Perhaps the author should consider replacing this with a flowchart.

5. I think the readers of this journal would mostly benefit from a well-explained example throughout. This could be perhaps the Lac Operon model from Figure 3. Using one example may better help readers understand the tool.

6. The idea of reaction schemas seems very interesting/compelling for this work! I would like to see this idea expanded and explained in a biological context more thoroughly. The authors instead start by defining the need for a schema but quickly devolve into mathematical and algorithmic details that likely belong in the supplement or in a more specialized section.

Minor comments:

1. Various spelling mistakes are present throuhgout. For example, in the abstract the authors wrote "complies" when I believe they meant "compile".

2. Similarly, the tone of the paper often sounds like a user manual or an advertisement for BioCRNpyler rather than a tool that solves a biological problem.

3. Figure 1 is too cluttered, while Figure 2 is not very informative. It is unclear to me what the other figures are trying to convey as well. The figures work best when they flow with the narrative.

4. Although the authors do a decent job in the introduction to present other tools, they also miss a chance to place BioCRNpyler in the context of other python tools. For example, Tellurium, COBRApy, COPASI, etc are Python tools that may be complementary to BioCRNpyler and should at least be mentioned.

**Have the authors made all data and (if applicable) computational code underlying the findings in their manuscript fully available?**

Reviewer #1: Yes

Reviewer #2: Yes

Reviewer #3: Yes

PLOS authors have the option to publish the peer review history of their article (what does this mean?). If published, this will include your full peer review and any attached files.

Reviewer #1: **Yes: **Chris Myers

Reviewer #2: No

Reviewer #3: No
---

## [Decision Letter · Decision Letter 1]

14 Feb 2022

Dear Mr. Poole,

Thank you very much for submitting your manuscript "BioCRNpyler: Compiling Chemical Reaction Networks from Biomolecular Parts in Diverse Contexts" for consideration at PLOS Computational Biology. As with all papers reviewed by the journal, your manuscript was reviewed by members of the editorial board and by several independent reviewers. The reviewers appreciated the attention to an important topic. Based on the reviews, we are likely to accept this manuscript for publication, providing that you modify the manuscript according to the review recommendations.

Sincerely,

Pedro Mendes, PhD

Associate Editor

PLOS Computational Biology

Dina Schneidman

Software Editor

PLOS Computational Biology

[LINK]

Reviewer's Responses to Questions

**Comments to the Authors:**

Reviewer #1: The authors have addressed all my concerns.

Reviewer #2: The authors significantly improved the readability of the manuscript.

My general note is about supplemental material. In Figures 4 and 6 it would be very useful to quickly look into the supplemental material and find a snippet of a code describing multiple models mentioned in the manuscript. However, there are no direct references to supplemental material describing specific code. The same way as the authors refer to specific supplemental tables, it would be nice to refer to specific sections of supplemental material describing models. Now a reader needs to read the whole supplemental material to match it to specific figures in the manuscript.

There are minor issues in the manuscript that should be fixed:

Lines 91-96 are almost verbatim repetition of lines 70-74. Perhaps the introduction can be without specific terms that would come later.

Line 175: exclamation sign is not necessary.

Lines 226-227: are “a promoter (prom), ribosome binding site (rbs)” necessary for the first code? Explain when and why you’ll need them, or remove – they confuse the reader.

Line 259: fix ComponentEnuemrators

Line 318: is DNAasembly the same as defined in line 234?

Figure 4: explicitly refer to each part of supplemental material for code snippets for each model.

Figure 5 – what do colors for LacOperon species (light blue, dark blue, bright blue and violet) mean?

Lines 424-425: “bio-modeling course and bootcamps with dozens of users ranging from college freshmen and sophomores with minimal coding experience“ - it reads like a grant application. I would remove it or be more specific on why these users use this software, e.g. which biological systems do they model that it makes it easier.

Link http://buildacell.io/BioCRNPyler from https://github.com/BuildACell/bioCRNpyler is dead.

Reviewer #3: The authors have addressed all my concerns with this revision. There are still some lingering typos throughout (e.g. complies -> compiles that should be fixed.

**Have the authors made all data and (if applicable) computational code underlying the findings in their manuscript fully available?**

Reviewer #1: Yes

Reviewer #2: Yes

Reviewer #3: Yes

PLOS authors have the option to publish the peer review history of their article (what does this mean?). If published, this will include your full peer review and any attached files.

Reviewer #1: **Yes: **Chris J. Myers

Reviewer #2: No

Reviewer #3: No

Figure Files:

Data Requirements:

Reproducibility:

References:

---

## [Editor Report · Decision Letter 2]

3 Mar 2022

Dear Mr. Poole,

We are pleased to inform you that your manuscript 'BioCRNpyler: Compiling Chemical Reaction Networks from Biomolecular Parts in Diverse Contexts' has been provisionally accepted for publication in PLOS Computational Biology.

Best regards,

Pedro Mendes, PhD

Associate Editor

PLOS Computational Biology

Dina Schneidman

Software Editor

PLOS Computational Biology

---

## [Editor Report · Acceptance letter]

12 Apr 2022

PCOMPBIOL-D-21-01367R2 

BioCRNpyler: Compiling Chemical Reaction Networks from Biomolecular Parts in Diverse Contexts

Dear Dr Poole,

I am pleased to inform you that your manuscript has been formally accepted for publication in PLOS Computational Biology. Your manuscript is now with our production department and you will be notified of the publication date in due course.

With kind regards,

Agnes Pap
